# Scalable Newborn Screening Solutions: Bioinformatics and Next-Generation Sequencing

**DOI:** 10.3390/ijns7040063

**Published:** 2021-09-30

**Authors:** Nicole Ruiz-Schultz, Bryce Asay, Andreas Rohrwasser

**Affiliations:** Utah Public Health Laboratory, Salt Lake City, UT 84129, USA; nruiz-schultz@utah.gov (N.R.-S.); aphlnbsba1@utah.gov (B.A.)

**Keywords:** next-generation sequencing, bioinformatics, genomic variants, newborn screening

## Abstract

Expansion of the newborn disorder panel requires the incorporation of new testing modalities. This is especially true for disorders lacking robust biomarkers for detection in primary screening methods and for disorders requiring genotyping or sequencing as a second-tier and/or diagnostic test. In this commentary, we discuss how next-generation sequencing (NGS) methods can be used as a secondary testing method in NBS. Additionally, we elaborate on the importance of genomic variant repositories for the annotation and interpretation of variants. Barriers to the incorporation of NGS and bioinformatics within NBS are discussed, and ideas for a regional bioinformatics model and shared variant repository are presented as potential solutions.

## 1. Current State of Variant Analysis in NBS

Genetic analyses in newborn screening (NBS) are typically performed as secondary screening methods or as part of the diagnostic process where analysis is limited to a set of variants, a single gene, or a set of genes associated with a particular NBS disorder. Second-tier testing in NBS is defined as orthogonal methods to perform testing on the same dried blood spot sample to increase specificity and reduce false-positive screening events. This is in contrast to diagnostic testing, which refers to the testing of an independently collected specimen, tested by a clinical reference laboratory employing a validated test and resulting in clinical management action. Genetic analysis assays most commonly used in NBS include genotyping panels and Sanger sequencing [1]. Next-generation sequencing (NGS) has recently been utilized in a small number of NBS programs in the US, primarily in the form of variant analysis panels for disorders such as cystic fibrosis (CF) (e.g., Illumina MiSeqDx CF assay) and severe combined immunodeficiency (SCID) (e.g., Archer Dx SCID panel), with the bioinformatics analysis either performed on the sequencing instrument or through a custom variant analysis pipeline on a high-performance computer [2]. A targeted panel approach for detection of disease-causing variants in disorders where genotype-phenotype associations are well characterized (e.g., CF, phenylketonuria, VLCAD deficiency) can be a more suitable approach in terms of lower costs and analysis restriction to prevent the identification of variants outside the gene(s) of interest with undetermined clinical consequences. However, for multi-gene disorders such as SCID or methylmalonic acidemia (MMA), where gene and variant associations are not well defined, a targeted approach can be less effective. With the increase in the number of disorders included on the Recommended Uniform Screening Panel (RUSP) and as NBS programs move towards the inclusion of emerging/rare disorders (e.g., X-linked adrenoleukodystrophy, Pompe disease, metachromatic leukodystrophy), targeted variant panel approaches can be incomplete as a secondary assay due to extremely limited variant knowledge and/or conflicting interpretations among laboratories and within the literature [3,4,5,6,7]. This will require the development of scalable testing methods that can effectively detect potentially clinically relevant variants in a population screening setting. Scalable testing methods are methodologies that would not require extensive redesign and revalidation when expanded to additional variants, genes, or disorders. Scalable methods also allow for additional analysis without the requirement of rerunning the laboratory portion of the test. In whole-exome sequencing (WES) or whole-genome sequencing (WGS) approaches, scalability allows the inclusion of additional genes in subsequent analyses after an initial restricted analysis does not detect any variants. In the event of a disorder added to the NBS panel, these methods would not require comprehensive assay revalidation. Only the bioinformatics portion of the assay would require revalidation. In contrast, amplicon-based methods require extensive revalidation of both the laboratory and the analysis methods. 

## 2. NGS Methods as a Way to Augment Secondary Variant Analysis in NBS

A universal solution to improve secondary variant analysis in NBS is defined by NGS-based testing methodologies that capture either part of or the entire genome, coupled with bioinformatics pipelines that restrict the analysis to a set of variants, a single gene, a set of genes, or can be expanded to the entire genome if clinically indicated. Examples of these methods include WES, WGS, and rapid WGS (rWGS). While WES is limited to the coding region of the genome, WGS has the advantage of detecting variants throughout the entire genome, including those located in deep intronic regions. rWGS allows faster full genome sequencing compared to traditional WGS due to lower overall genome coverage. These types of analysis methods provide multiple benefits to both the newborn and the NBS program, including fast turnaround time, the development and utilization of a single method that covers all disorders, reduction of false positives/referrals to diagnostic testing, and access to NGS tests for all newborns regardless of health insurance status. 

Since only the computational analysis and not the laboratory assay needs to be repeated when additional variants or genes need to be included in expanded analyses, this type of testing platform is ideally suited for this task. Time to diagnosis is an important consideration for NBS programs utilizing commercial testing solutions. Turnaround time for many genetic testing companies to complete testing ranges from 3–6 weeks, often far too long for time-critical disorders [8]. For example, for infantile-onset Pompe disease (IOPD), treatment should be initiated as soon as possible, ideally within two weeks from the time that IOPD is suspected. If NGS testing is performed in-house by NBS programs, it is likely that such timeliness requirements would be met.

NGS-based assays allow a program to add multiple disorders to their panel with the development and validation of a single assay. Within NBS, it is still common practice that one platform or assay is utilized for one single test or disorder. Examples of such platforms include the EnLite Neonatal TREC instrument for SCID screening and the Luminex xTAG assays for CF. In contrast, NGS-based assays provide the scalability that current variant analysis methods do not afford. An example of this is the pipeline the Utah NBS program has developed. In this approach, we use WES with custom bioinformatics pipelines that restrict the analysis to a single gene initially, but any analysis is expandable to the entire exome if sequencing coverage suffices [9]. This pipeline was used in a pilot study as a second-tier screening method to confirm the results of biochemical screening for metachromatic leukodystrophy (MLD) [10]. Our NGS method was able to identify clinically significant variants for two screen-positive samples, thus improving the specificity for the biochemical MLD screening method. Our program has recently implemented screening for X-ALD and will soon begin screening for PD as well. A benefit of our NGS assay is that it allows for secondary screening of samples that are abnormal and borderline abnormal, which can help in making final determinations regarding referral to diagnostic testing as well as in the cut-off verification process for the first-tier biochemical screening method. 

Benefits of NGS testing in NBS also include universal access to testing that might otherwise be out of reach for families due to a lack of insurance or their insurance not covering certain tests. Today, many programs include second-tier testing as well as diagnostic testing in their respective NBS systems and workflows. With NGS testing becoming a standard secondary test in NBS, cost barriers are eliminated for all families. While this would reduce diagnostic odysseys, it would also reduce delays as the result of administrative delays often associated with seeking payment authorization from insurance providers. 

The potential utility for the use of NGS in NBS has been explored through various pilot studies [11,12,13,14,15,16]. Some of these studies were funded and overseen through the Newborn Sequencing in Genomic Medicine and Public Health (NSIGHT) Consortium. Established in 2013, the goals of this six-year program were to determine if genome sequencing could replicate and/or improve upon current NBS methods, provide additional information about conditions not on the NBS panel, and provide additional information that could be used in clinical decision making for newborns [17]. The NSIGHT Consortium projects showed that genome sequencing can be a useful diagnostic tool for NICU patients, can complement NBS methods, and can be a good second-tier test for NBS [18]. However, before we can consider employing NGS on a large scale in NBS, standardized bioinformatics tools, pipelines, frameworks, and resources should be developed and shared with the community.

## 3. Genomic Variant Repositories—A Key Tool in Variant Interpretation

Variant interpretation, a key component of the genomic variant bioinformatics pipeline where the clinical impact of a variant is determined based on precedence, relies on knowledge from various public genomic variant repositories as well as private, program-specific variant repositories. As stated previously, the level of variant curation for NBS disorders varies widely. In our recent publication, we compared variants associated with SCID, various metabolic disorders, and MLD that we collected and curated from ClinVar and multiple Leiden Open Variation Databases (LOVDs) [9]. Our results showed varying degrees of overlap between the databases for each gene/disorder (SCID: 3–31%, metabolic disorders: 23–65%, MLD: 9–36%) as well as conflicts with variant representation in Human Genome Variation Society (HGVS) format (a structured nomenclature used to describe genomic variants with regards to their position on the genome and their variant type), which resulted in 11% of the entries failing validation checks. The use of NGS methods in NBS allows for population-level screening of genomic variants associated with NBS disorders that will add to the knowledge of well-characterized disorders and aid in the efforts of interpretation of variants from disorders with limited information regarding genotype/phenotype correlation. Instead of siloing this data within private or program-specific databases, the NBS community must begin aggregating variant information (variant, clinical interpretation, zygosity, population allele frequencies) with observed biochemical phenotypes within repositories that can be shared across all NBS programs. NBS programs would be generating this data as well as using it within their interpretation pipelines. 

To capture and share variant information across all NBS programs, repositories and data sharing frameworks need to be created through which data can be collected, curated, and shared across NBS programs. Examples of these repositories exist within the rare disease community, such as CFTR2, the ALD mutation database, and BioPKU [19,20,21]. The value of a variant repository can be illustrated, with CF and the CFTR2 database as an example. Over time, as a community, we began to identify and understand the common variants associated with CF, such as the deltaF508 variant. With the introduction of population screening, the variant space expanded and covered increasingly rare variants. In turn, this knowledge led to the understanding of genotype/phenotype relationships, which, in turn, resulted in successful intervention protocols as well as the development of genotype-specific drugs [22,23]. 

With broad adoption of NGS-based variant analysis, we would similarly expect to classify a large number of variants as variants of unknown significance (VUS), but, over time, these interpretations would change as more data pertaining to the variant’s frequency and associated phenotypes are gathered. Again, we have seen this with CF, where the discovery of CFTR variants and their associated phenotypes led to a refined understanding of the disease. We expect this will hold true for many rare disorders as more of these disorders are included in NBS panels. We expand from a limited understanding of common variants of these disorders to the discovery and understanding of the rarer variants and their contributions to disease pathology. Through this, we gain an understanding of the disorder, and, eventually, the number of variants classified as VUS decreases.

Up to this point, there have not been efforts towards the centralized curation and sharing of genomic variants within the NBS community. One of such efforts that aim to address this is the CDC’s Enhancing Data-driven Disease Detection in Newborns (ED3N) tool [24]. Centralized repositories will need to meet certain requirements to be useful to all NBS programs (Figure 1). First, these systems will need to account for the different secondary molecular testing methods (Sanger, variant panels, gene panels, in-house NGS, third-party NGS) used by each state. The repository would need to be flexible, accommodating data interpreted from various assay types and integrating this information to make it comparable across different testing modalities. For example, each method has its own set of quality control parameters that should be included with any data submission. Additionally, a centralized variant repository would allow for version control and standardization of variant data. A quality control pipeline can be created to check data before it is entered into the repository. This pipeline could verify the HGVS annotation and transform the data into a standard format for the repository. Records regarding variant interpretation changes and version-controlled variant repositories would not only track changes but also alert users regarding the amendment requirements of any NBS or diagnostic reports. Different degrees of participation in a centralized repository should be allowed. Some programs may choose to only pull data from the repository but not submit data, while others may submit data and pull data from the repository. NBS programs might be willing to submit data but might be prohibited by their states’ statutory requirements from sharing health data even if in a de-identified format. States that are prohibited from sharing data should not be excluded from being able to use the repository. With regards to international variant data sharing, legislation such as the General Data Protection Regulation (GDPR), a set of standards that detail how companies should handle data of EU citizens to ensure personal data is protected, could be a barrier that prevents international participation in such an initiative [25,26]. 

An NBS-specific centralized repository should also have the ability to view conflicts in variant interpretation across contributors to enable the submitters to review the interpretation and to ultimately achieve consensus decisions. This is similar to ClinVar Miner as a way to mitigate conflict for ClinVar [27]. ClinVar Miner is a tool that renders data from ClinVar in a way that allows us to explore variant interpretation conflict as well as view statistics on variant data submission. One of the use cases for ClinVar Miner is to facilitate the ability to address variant interpretation conflict between all submitters of a single variant and come to a consensus interpretation.

Alternatively, disease-specific variant repositories created, hosted, and maintained by rare disease foundations could constitute intermediate or long-term solutions. Numerous foundations have shown agility and flexibility towards implementation and broad adoption. The CF Foundation maintains the CFTR2 database, which aggregates biochemical, variant, and other phenotype data of CF patients as well as provides a tool for healthcare providers that aids in variant interpretation [28]. Hunter’s Hope, a leukodystrophy-specific foundation, has created the Hunter’s Hope Krabbe family database to collect clinical data on leukodystrophy patients [29]. Considering that comprehensive phenotype–genotype relationships and characterization thereof will ultimately define broad utility, such decentralized options hosted by the rare disease community will also allow the leveraging of this knowledge towards targeted drug development efforts. Lastly, efforts led by the rare disease community would allow worldwide organization and curation, efforts that are an obvious benefit in the rare disease space.

## 4. Additional Challenges to Implementation

While NGS is a powerful tool for secondary molecular screening, there remain many challenges regarding implementation at the individual state or program level. The expertise required to establish and run a bioinformatics core does not currently exist in the majority of programs. Additionally, the ability to recruit and retain this expertise or talent in-house is hampered by the fact that most programs cannot afford the salaries that are offered by the private industry sector. There are also significant upfront costs to the purchase of sequencing instruments, reagents, and computational equipment necessary to perform sequencing within the NBS program. If these challenges cannot be overcome, it may drive some programs to adopt a solution where their secondary molecular analysis is performed by a third-party laboratory, such as a major genetic testing company. While this may seem like a good solution, using a private model for sequencing, variant databases, and bioinformatics analysis in NBS has potential negative impacts on the community. Most of the tests offered by these companies are panel-based, but NGS methods like WES or WGS are used. This allows the private entity to retain ownership of the data for NGS data mining and secondary R&D efforts, a resource that is lost to the individual NBS programs and the public at large. Technology and data that are kept from the public realm due to patents and costly subscriptions can stifle innovation. These barriers would need to be removed to allow for competition and to drive innovation within the field. 

Despite many NGS adoption barriers, it is feasible to introduce NGS technology in the NBS community. As described in Section 2, efforts of pilot programs have led to the applicability of NGS in NICU settings with positive impacts on clinical care. Several hospital systems are in the process of adopting this strategy [11,12,13,14,15,16]. While the NSIGHT Consortium funded self-standing projects and explored the feasibility of sequencing in NBS, we anticipate that when NBS programs adopt sequencing, there will be a common interest in improving the outcome of the methodology. With community interest and adoption, stakeholder support as well as program and policy support from federal partners (i.e., ED3N) should follow.

## 5. Potential Solution—A NBS Regional Bioinformatics Model

Regional organization, development, and implementation efforts are frequently referenced as solutions to address the problems of lack of bioinformatics expertise within NBS programs combined with long-implementation cycles. Figure 2 shows three potential ways a regional model could be implemented within the NBS community.

These models could also be combined into a hybrid model, depending on the needs of each individual program. The first model (A) demonstrates how NBS programs that do not have sequencing or bioinformatics resources can send specimens to NBS programs that act as a genomic sequencing hub where sequencing and analysis would occur. The interpreted and raw sequencing results would then be sent back to the program via Fast Healthcare Interoperability Resources (FHIR). Such a model would allow variant data sharing in an NBS variant repository. In the second model (B), NBS programs would perform sequencing in-house. Due to a lack of bioinformatics analysis capability, analysis and interpretation would be provided by a regional bioinformatics hub. Transmission of outbound raw sequencing outputs and inbound receiving of result files would be achieved through FHIR protocols. Such a model would, again, allow data sharing with a variant repository. In the third model (C), NBS programs are fully capable of generating and analyzing sequence data and would share variant data with the NBS variant repository. Regionalized efforts would not only address such structural barriers but would also provide economies of scale to address talent recruitment and retention regarding bioinformaticians and board-certified molecular geneticists reviewing screening and diagnostic results.

Technically, funding agencies could provide funding to focus on building such resources in terms of both technical infrastructure as well as talent recruitment and retention. Specific grants could target the purchase of standardized equipment and analysis infrastructure. Adoption of sequencing and bioinformatics technologies and resources in public health’s infectious disease management efforts have not only shown feasibility but have also demonstrated high success probabilities and rapid adoption using regionalized coordination and resources. Examples of these programs include the Antibiotic Resistance Laboratory Network (AR Lab Network) and PulseNet. The AR Lab Network is a CDC laboratory network established in 2016 that provides the ability to rapidly detect, track, and respond to antibiotic-resistant pathogens [30]. This network includes laboratories throughout the US as well as seven regional laboratories that provide support to smaller state and local level laboratories. PulseNet USA is another CDC laboratory network established in 1996 that functions as a surveillance system to rapidly detect outbreaks caused by foodborne bacterial pathogens [31,32]. An example within NBS is the Collaborative Laboratory Integrated Reports (CLIR) tool, which began in 2004 as a network of laboratories throughout the world to improve tandem mass spectrometry and biochemical NBS and continues to successfully serve the NBS community [33]. Furthermore, regionalized screening resources through renowned medical centers and reference laboratories show feasibility and success potential.

Innovative solutions could entail public/private partnerships directly or indirectly, establishing sequencing-based diagnostics or bioinformatics resources in NBS. Such efforts could range from directly or indirectly providing diagnostic resources as part of diagnostic workflows, working with centralized public health associations on pricing and testing resource plans, and working with disease foundations to host or fund such infrastructures to including pharmaceutical partners working with diagnostic service providers, establishing diagnostic services and, at the same time, ensuring broad data sharing. Rare disease foundations have shown interest in establishing these types of resources as a number of these groups have created and currently maintain repositories for variant and/or other clinical data (Section 3). Perhaps an additional role for these foundations would be to collaborate with NBS programs to enhance the repositories by accepting additional data types (i.e., biochemical test results, genomic variant results) from NBS programs, and, in turn, NBS programs can use data from these foundations for algorithm improvement or variant interpretation. Alternatively, collaborations with pharmaceutical partners could be a way for NBS programs and rare disease foundations to fund and collaborate on building repositories and connect the community.

## 6. Driver for More Rapid Adoption

The rapid emergence of highly promising treatment modalities for rare and ultra-rare disorders that would benefit from NBS and diagnosis prior to the onset of symptoms emphasizes the need for scalable screening and diagnostic methodologies. Successful and rapid implementation of sequencing and bioinformatics tools as one of such scalable methodologies stresses the need for programmatic development efforts and accelerated implementation. This need also emphasizes that the current NBS system is not positioned to respond quickly and requires structural adaptation and system changes. However, worldwide successes of the rare disease community, ranging from foundations to individual families advocating the rapid expansion of NBS, and the inclusion of more and more disorders into NBS panels while stressing current public health efforts will ultimately drive the innovation and adoption of scalable methods and novel mechanisms of integrating genomic diagnostics in NBS practice.

## Figures and Tables

**Figure 1 IJNS-07-00063-f001:**
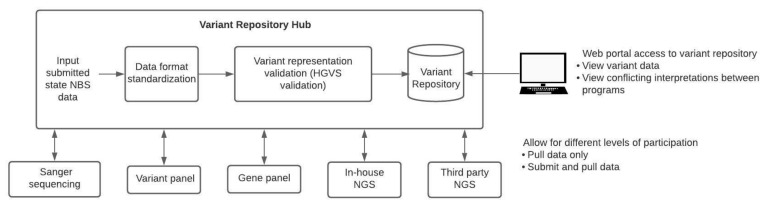
Curation and sharing of variant data across NBS programs. To aggregate and share NBS variant data across programs, a pipeline for data standardization is required. State NBS programs would be able to submit data to a centralized variant repository hub that would standardize the various data formats and variant representations before aggregating this data in a variant repository. Multiple levels of participation would need to be allowed since some states may choose to submit and pull from the repository, whereas other states may wish to only pull data from the repository. Web portal access to the variant repository would allow bioinformaticians as well as non-bioinformaticians to access and view variant data as well as view conflicting interpretations between programs.

**Figure 2 IJNS-07-00063-f002:**
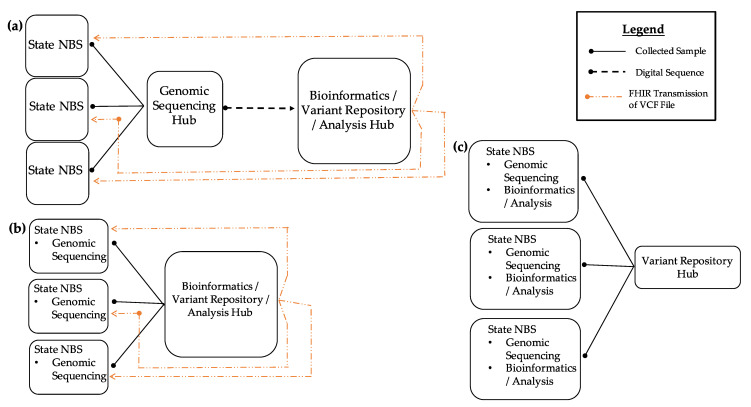
Regional bioinformatics model in NBS. (**a**) In the first model, states lacking a sequencing/bioinformatics program can send their specimens to an NBS program or a diagnostic testing center that acts as a genomic sequencing/bioinformatics hub. Results can be returned via Fast Healthcare Interoperability Resources (FHIR) transmission of VCF files. (**b**) In the second model, states may have the ability to perform genomic sequencing in their individual labs but lack bioinformatics expertise for analysis. These states can send their sequence data to a state NBS program acting as a bioinformatics hub for analysis and have the results returned via FHIR. (**c**) In the third model, all states have sequencing and bioinformatics analysis resources. In all models, state NBS programs are able to share genomic variant data with the entire NBS community.

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
