# Peer review of "Scalable Newborn Screening Solutions: Bioinformatics and Next-Generation Sequencing"

_2409-515X, 2021, doi:10.3390/ijns7040063_

Round 1

Reviewer 1 Report

This is  commentary and as such not a research paper, which is why some of the answers above have been written as NA.

Authors describe that NGS as secondary testing method in NBS and that more extensive use of NGS could fulfill the emergent need for a scalable  screening. Argues for NGS based testing of part of or entire genome with a powerful bioinformatics pipeline to narrow down the genes of interest - this would give increased scalability. The importance of genomic variant repositories is stressed and there is not a sufficient infrastructure for this yet. Along that lines the authors suggests three models for a regional bioinformatics model and shared variant repository. Variant information together with biochemical information to be shared across NBS programs. A number of  practical problems associated with this are pinpointed and relevantly discussed. Legal aspects are touched upon - one could wish the  authors would also mention the GDPR legislation, which is an obstacle for some European countries to join the CLIR-instrument and they would also have problem joining the proposed repository. I think the work is sound and pinpiont a very important point about the eminent introduction of NGS in NBS. Discussion and references are relevant.

Langauge is excellent. One could discuss the use of the word "polygenic" for diseases like SCID and MMA in section 1 - though many genes are involved in these diseases, I would not normally call them polygenic (as for parkinson or hypertension). Two "that"s are noted in last paragraph of section 3.

Author Response

Thank you very much for your positive and constructive feedback. We have made the changes you have suggested and believe we have an improved version thanks to you. Specifically, this is how we addressed each of your suggestions:

  1. We have added a sentence (lines 180-184) describing how the General Data Protection Regulation (GDPR) would make it difficult for European countries to participate in a variant data sharing initiative. 
  2. We have replaced polygenic with multi-gene when mentioning SCID and MMA in line 36.
  3. We have removed the additional “that” from the last paragraph of section 3.

Reviewer 2 Report

This commentary in IJNS is appropriate and timely. The US healthcare system is different than in Canada and Europe, but nevertheless implements productive and innovative solutions in healthcare. The pillars of healthcare are testing, drugs (made by American Biopharmaceutical companies), and the healthcare providers that deliver them along with surgical procedures. In testing, rare genetic disease testing is complicated by the reimbursement process. Since 2010 Exome sequencing has made great strides in Mendelian Genetics Research and making contributions in diagnostics. Newborn Screening is a sure shot way to implement the testing that resolves a gap in the healthcare delivery model and reduces disparity. As new drugs come down the pipeline like in Pompe disease or Duchenne, the utilization of these NGS tools may be valuable to newborn screening undertaken by public health laboratories. I strongly support the efforts this team is putting in Utah, and ultimately the value this provides to the US NBS infrastructure and the exemplar benefit of that worldwide.

  1. Some suggestions:
    1. Title: Bioinformatics and Next-Generation Sequencing Implementation in Newborn Screening. The title of scalability is not really discussed and may be confusing or non-specific.
    2. ABSTRACT: lines 7-9; As additional disorders are added to the newborn screening (NBS) panel, this necessitates the incorporation of new testing modalities. This is especially true for disorders lacking a robust biomarker for detection in primary testing methods, and for disorders requiring genotyping or sequencing as a second-tier and/or diagnostic test. In this commentary, we discuss how next-generation sequencing (NGS) methods can be used as a secondary testing method in NBS. Comment: Disorders are added because of new treatment in the pipeline and early onset. The testing may not be added to the panel if the test does not exist or utility demonstrated via a pilot. NGS provides a scalable solution. Lacking a biomarker makes genetics an orthogonal approach, and the second tier is mostly to reduce FP of the first-tier test. The authors should discuss these in the manuscript or modify the comments.
      1. Line 32-34: The words polygenic and targeted approach is confusing. The authors should consider heterogeneous disorder or multi gene disorders that more than one gene. The authors may be trying to say as the heterogeneity increases and panel size increases or there are modifiers etc. that contribute to disease severity and thus the utility of a panel or genotyping approach becomes limited. The authors in lines 39-40 do talk about variant interp or variant effects prediction, but this will require some references. However, this challenge is different from scalability, and scalability in light of these discussions is difficult to understand. So the authors must define what they mean by scalability in a short sentence or two so readers are able to follow.
    3. Section 2: NGS Methods as a Way to Augment Secondary Variant Analysis in NBS. Comment: The section is weak in the reviewer's opinion. The authors do not outline why the community should consider employing NGS on scale in NBS rather than disease-specific secondary biochemical testing. The authors start out in the introduction section talking about the diseases but they do not outline that rationale here. The value of genome-scale sequencing as a need is not outlined. Most of what is written here is what can be done, but why it should be done is not described. Lines 53 should be modified to highlight the lessons learned from Utah WES programs (if any) to support the claims. Many PHL in this country routinely do send-outs to private labs for second-tier testing or secondary testing. One example is the approach described by Smith et al., in IJNS (2020) and Ficicioglu (2020) for Pompe. The lack of insurance for many families (on Medicaid) forces and makes it impossible to get confirmatory testing and needed therapy in time especially for CRIM-ve status in Pompe where the timely diagnosis is important. However, what many state labs end up doing is sending it for a second-tier biochemical test due to VUS or cost issues. Unfortunately, though unproven, these biochem second-tier tests and similar types in other diseases will never help the families if they have infantile onset diseases that require immunomodulation therapy or a specific therapy (Pompe infantile condition or PKU biopterin defects etc.). This is never raised as an issue in Pompe disease by the community, because the cases are rare and far in-between, but overall this responsibility is dumped on the diagnostic team that has to go through authorization etc. This type of example may be useful to write by the group for NBS because the responsibility is not just sorting FP but making a difference in the clinical algorithm and its modernization and implementation.
    4. Also pilot studies argument is unclear (lines 60-62). To some extent pilots are needed as demonstration projects and getting people alignment, but it is true that it is not sufficient. So this needs to be restated.
    5. Section 3: Genomic Variant Repositories: Is the variant framework clear? It is vague to understand how this will be done. CFTR and PKU would be good examples for the group to describe in more detail. Best would be to talk to the CF and PKU clinicians and come up with what the value would be. In CF the disease is a spectrum (CRMS) disorder and many cases are not CF. Is there a lesson to be learned. In PKU the treatment needs to be understood. The variant repository value is not clear…… (line 84-125). When you look at genes the number of variants discovered will increase as more babies are sequenced. Many of these would be VUSes just like the BRCA genes sequenced by Myriad in early days were VUSes. In the long run the genoype-phenotype will be stronger and this will help cascade testing, at risk testing and f/u.
    6. Line 73. Not clear to general audience. Very technical term.
    7. Line 74- important concept…NBS databases really do not exist. Most are older patients or pediatric patients. A newborn cohort of normal and disease does not exist either. Knowing cohort level MAF s or allele frequencies, biochem values etc. are important (like pseudo deficiencies in Pompe or LSDs).
    8. Lines 100-116. A diagram here could help.
    9. Additional Challenges Section: Lines 127-142. The issue here is also who would drive this, how will consensus happen etc. how would partnerships work and how would there be pure benefit driven algorithm improvement given many stakeholders? NBS has the potential to allow expensive tests that are heritable disorders be addressed upfront so damage is halted and newer treatment options available and utilized. It is the most important thing to do instead of the broken symptomatic testing which is like a post-mortem testing masquerading as a confirmatory test. In most cases when overt symptoms are recognized, significant damage has occurred, so you want to know as early as possible. This model has not been embraced despite many projects like the U19 (NSIGHT) on Newborn Sequencing between 2012 and now (Smith et al. 2020).
    10. For section 5 and 6. The reviewer recommends the suggestions be made more specific, with examples etc. as otherwise it is too difficult to conceptualize and is abstract. Like foundations: what would their role be. Will they be interested in funding? Most of the time they give out ad hoc research funds for drug discovery or to a PI for research. They rarely give money for infrastructure. Most money from NIH sources are for innovative research. Clearly this is a major problem and a streamlined investment stakeholder approach could be very useful here. There should be both public and private and collaborative opportunities.
    11. Lines 100-116. Implementation does not talk about why a private model will or will not work.
    12. Long term sustainability is also not clear (lines 173).

Round 2

Reviewer 2 Report

This looks good. Changes are made are appropriate.